# Moving Northwards: Life-History Traits of the Invasive Green Crab (*Carcinus maenas*) Expanding into the Southwestern Atlantic

**DOI:** 10.3390/biology14050480

**Published:** 2025-04-26

**Authors:** Micaela Müller Baigorria, Maite Narvarte, Leandro A. Hünicken

**Affiliations:** 1Genética y Ecología Evolutiva, Centro de Recursos Renovables de la Zona Semiárida (CERZOS, CONICET-UNS), Camino La Carrindanga Km 7, Bahía Blanca 8000, Argentina; micaela.muller431@gmail.com; 2Facultad de Ciencias Marinas, Universidad Nacional del Comahue, San Martin 224, San Antonio Oeste 8520, Argentina; 3Centro de Investigación Aplicada y Transferencia Tecnológica en Recursos Marinos (CIMAS, CONICET-UNCo), Güemes 1030, San Antonio Oeste 8520, Argentina; mnarvarte@gmail.com

**Keywords:** *Carcinus maenas*, life-history traits, sex ratio, population density, juvenile abundance, ovigerous females, newly established population, expansion range

## Abstract

*Carcinus maenas* is an invasive crab species that threatens coastal ecosystems around the world by competing with native species and damaging habitats. Recently, a population of this species was discovered in the San Matías Gulf in Argentina, marking its northernmost record in the southwestern Atlantic. We studied its demographic and life-history traits in the intertidal zone and found a male-biased sex ratio and relatively low population density. Additionally, egg-bearing females were present nearly year-round, with an increase in their abundance from May to July, suggesting that this population is adapting well to local conditions and maintaining an extended reproductive period. Together, these findings point to an early stage of invasion with the potential for further expansion, posing risks to native species and north Patagonian ecosystems.

## 1. Introduction

The establishment and expansion of invasive species depend on multiple factors, including their biological traits and the environmental conditions of the invaded region [1]. In newly established populations, individuals often show high growth rates and an extended reproductive period, traits that facilitate population expansion [2]. In marine ecosystems, invasive crustaceans significantly alter native biodiversity and ecosystem functions [3]. The European green crab *Carcinus maenas*, one of the world’s most notorious marine invaders, has successfully spread across multiple continents due to its adaptability and high reproductive potential [4,5,6,7].

The population dynamics of *C. maenas* are shaped by its ability to rapidly colonize new environments. These crabs exhibit flexible growth patterns, with rates varying depending on environmental conditions such as temperature, food availability, and population density [5]. Additionally, *C. maenas* have been observed to shift reproductive strategies in response to low densities, such as earlier maturation and prolonged breeding seasons, which enhance invasion success [8].

The green crab exhibits remarkable tolerance to diverse environmental conditions. Adults can withstand temperatures ranging from 0 to 33 °C [5], while optimal larval development occurs within a narrower range of 10 to 22.5 °C [9]. This species also endures a wide salinity range: early life stages require salinities above 20‰ [10], whereas adults survive in salinities from 10 to 50‰ [11]. Temperature plays a key role in range expansion, influencing growth and maximum body size [12]. Larger individuals tend to have greater reproductive success, fecundity, and longevity. In females, increased body size correlates with a higher number of eggs per brood, boosting larval output and enhancing establishment success [13]. Additionally, extended longevity prolongs the reproductive period, allowing for more reproductive events over an individual’s lifetime [12].

The species was first recorded outside its native range along the mid-Atlantic coast of the United States in 1817, and it has since expanded globally, facilitated by international trade and ballast water transport [8]. Established populations now exist in Asia, Australia, and South Africa, among other regions (references in [5]). In the southwestern Atlantic, *C*. *maenas* was first reported in 2001 in Bahía Camarones, Chubut province, Argentina [14,15]. It has steadily expanded its range along the coast, reaching Puerto Madryn by 2015 [16]. In 2019, Malvé et al. [17] documented an individual in Puerto Lobos, suggesting a potential incursion into the San Matías Gulf. By 2022, a population was confirmed in the northern region of the Gulf [18] (Figure 1), marking the northernmost distribution of *C*. *maenas* in the southwestern Atlantic.

This progressive northward expansion in the southwestern Atlantic could generate significant changes in the trophic networks of coastal ecosystems in southern South America [19] and jeopardize the livelihoods of shellfish-based fisheries, since this crab primarily preys upon mollusks and crustaceans in coastal ecosystems [20]. Despite the ecological and economic implications, there remains limited knowledge about the demographic and biological traits of the invasive green crab at the northernmost distribution in the southwestern Atlantic. Understanding the demographic and life-history characteristics of these crabs is a key aspect for unraveling the mechanisms behind their range expansion.

Here, we aim to provide baseline demographic and life-history data for the newly established population of green crabs at the northernmost edge of its known range in the southwestern Atlantic. Specifically, we assess key population parameters, including population density, size distribution, sex ratio, and the proportion of ovigerous females.

**Figure 1 biology-14-00480-f001:**
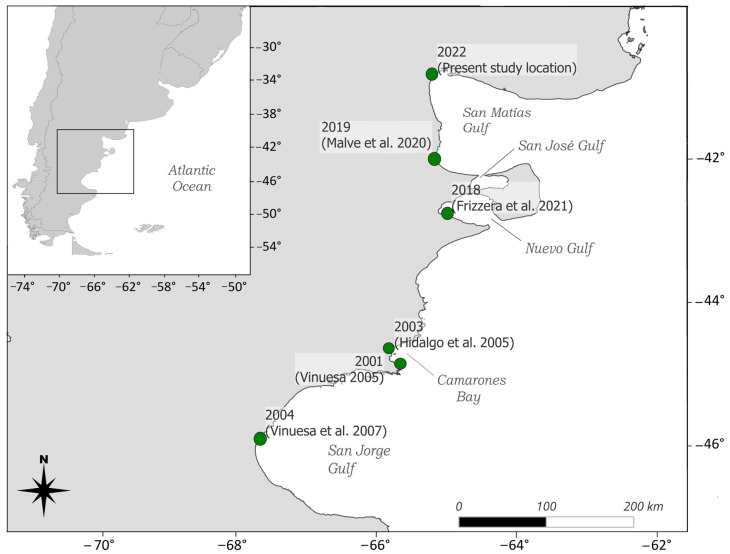
Distribution of the invasive green crab *Carcinus maenas* along the southwestern Atlantic coast. Green dots represent previously documented populations, the year of observation, and corresponding citation indicated [14,15,17,21,22].

## 2. Materials and Methods

### 2.1. Study Area

This study was conducted in the rocky intertidal zone of northern San Matías Gulf, Río Negro Province, Argentina (40°49.88′ S, 65°6.46′ W). Water temperatures range from an average of 9.8 °C in July to 21.4 °C in December (Servicio de Hidrografía Naval, https://www.hidro.gov.ar (accessed on 20 November 2024)), while salinity varies from 33.94 ± 0.5‰ in April to 34.03 ± 0.7‰ in September [23]. The intertidal zone extends for over 200 m across the tidal gradient. Its topography is characterized by a gently sloping ramp on the seaward side, influenced by a semidiurnal macro-tidal regime with up to 9 m of amplitude and significant seasonal variation [24]. Close to the mean high tide level, the upper intertidal area is dominated by macroalgae and tide pools. During low tide, these upper areas experience high desiccation and fluctuations in temperature and salinity, as the tide takes approximately 12 h to cover them [24]. Low-lying areas near the seaward edge are characterized by dense populations of the mytilids *Brachidontes rodriguezii* and *Perumitylus purpuratus*, as well as barnacles. The tidal cycle exposes organisms to extreme yet predictable changes in abiotic conditions twice daily, likely influencing their vertical distribution and zonation patterns. Consequently, the intertidal zone can be characterized into two distinct, but spatially adjacent, areas.

### 2.2. Field Sampling

We conducted monthly sampling of green crabs from August 2022 to July 2023 in the intertidal zone of San Matías Gulf. Sampling efforts focused on both the upper and lower intertidal areas, with a simple random sampling design used to allocate quadrats within each area. During low tide, we laid out four 25 m^2^ quadrats per area to obtain a representative sample and estimate population density.

We manually collected crabs by performing an exhaustive search under rocks and among algae, using metal hooks to probe inside burrows. Non-destructive methods were prioritized to minimize the impact on the population and habitat. To estimate individual size, we measured the carapace width (CW) to the nearest 0.1 mm using a digital caliper, recording the distance between the distal ends of the fifth teeth on the anterolateral margin of the carapace. Crabs were sexed through a visual inspection of the abdomen (Figure A1), and small individuals that could not be sexed were categorized as juveniles. Additionally, we recorded the presence of ovigerous females.

Following measurements and observations, subsamples were transported to the laboratory for later species identification while the remaining specimens were released back into their habitat. These samples were preserved in the Invertebrate Collection of CENPAT (Centro Nacional Patagónico, cataloged under the numbers CNP-INV 4113 and CNP-INV 4116). The procedure for identifying these individuals is described in the “Species Identification” section of Appendix B.

### 2.3. Statistical Analyses

All statistical analyses were conducted using R version 4.3.1. To analyze green crab abundance (density) and size (carapace width), we fitted a generalized linear mixed model (GLMM) and linear mixed model (LMM), respectively, with the lme4 package [25]. For green crab density, we used a GLMM with a negative binomial distribution and log-link function. We included log (25 m^2^) as an offset term to convert count data to density values (ind m^−2^), accounting for the sampling area in each plot. Both models included sex (male and female) as a fixed factor, while month (sampling dates from August 2022 to July 2023) and intertidal area (upper and lower) were incorporated as random effects to account for temporal and spatial variability. The model selection involved testing the significance of fixed and random effects using likelihood ratio tests (LRTs), comparing models with and without each factor while retaining other effects. Random effects were tested following the corrections outlined by [26].

To analyze the population size structure of the crabs and identify distinct size groups, we applied a finite mixture model approach. We implemented the mixture model using the mixtools package [27], utilizing the Expectation-Maximization (EM) algorithm [28] for parameter estimation. Models with one to five components were fitted for each sex, with the optimal number of components determined by the Bayesian Information Criterion (BIC; [29]). We used k-means clustering to enhance EM algorithm convergence [30]. A Separation Index (SI) was calculated to assess the distinctness of adjacent components, and components with SI < 2 were merged to prevent overfitting [31]. Monthly analyses were conducted to investigate seasonal patterns and growth progression.

To estimate the size at which 50% of female crabs are ovigerous (SM_50_; [16,32,33]), we grouped female individuals into carapace width (CW) intervals of 5.3 mm, determined using Sturges’ formula for class intervals [34]. We applied a weighted logistic regression using a generalized linear model (GLM) with a binomial error distribution and a logit link function. The binomial response variable was defined by the presence and absence of ovigerous females, with the carapace width (CW) as the independent variable. We applied weights corresponding to the total number of individuals in each size class to adjust for differences in sample sizes across classes, ensuring that size classes with larger sample sizes contributed proportionally more to the model.

Additionally, we calculated the monthly proportion of ovigerous females (number of ovigerous females divided by total females examined each month) and further computed the distribution of reproductive effort throughout the year by expressing each monthly count of ovigerous females as a fraction of the annual total. To analyze the degree of seasonality in reproductive patterns, we calculated the coefficient of variation (CV) using these weighted annual proportions.

## 3. Results

### 3.1. Density and Individual Size

A total of 1300 crabs were collected, and we found that males outnumbered females, yielding a sex ratio of 1.50 (709 males and 472 females). The overall density of green crabs in the San Matías Gulf was 0.42 ± 0.05 ind m^−2^; consistently, males showed, on average, a 79% higher density compared to females (Table 1). The lowest density for both sexes was observed in August, in the winter season, followed by a sustained increase in density during the warmer months from December to February (Figure 2). Temporal and spatial factors exerted greater influence on crab density than sex. The differences between males and females, although consistent throughout the year and between zones, explained only 3.7% of the variability (marginal R^2^, Table A2). Conversely, random effects accounted for 26% of the total variance, with similar contributions from month and zone (14.3% and 12.1%, respectively; Table A2), resulting in the complete model explaining 28.6% of observed variability. The juvenile density was 0.028 ± 0.003 ind m^−2^.

We also found a significant difference in size (carapace width, CW) between male and female individuals. On average, males (40.11 mm) were 9% larger than females (36.79 mm; Table 1). The CW range was 16.4–64 mm for females and 11.0–72.1 mm for males. Juveniles showed, on average, a CW = 12.4 ± 0.4 mm (4.5–18.9 mm).

### 3.2. Population Size Structure

The mixture model analysis identified two distinct size classes for both male and female green crabs in the overall population (Figure A3). For males, these size classes were characterized by means of 27.13 ± 7.13 mm and 48.72 ± 8.95 mm, comprising 37.24% and 62.76% of the male population, respectively. Female crabs similarly exhibited two size classes with means of 25.54 ± 5.34 mm and 40.49 ± 7.81 mm, representing 20.13% and 79.87% of the female population, respectively.

We observed a considerable temporal variation in the size class distribution throughout the annual cycle (Figure 3). Male crabs predominantly displayed two size classes throughout the year, with the exception of August, when three modes were observed. The smaller male size class (young adults) showed mean carapace widths ranging from 15.87 mm (August) to 36.86 mm (January), while the larger size class ranged from 22.40 mm (August) to 58.30 mm (January). Females exhibited a more variable modal structure, ranging from one to three size classes depending on the month. The smaller female size class ranged from 20.28 mm (February) to 39.55 mm (May), while the larger size class ranged from 38.52 mm (June) to 55.59 mm (January).

The primary finding was the emergence of a substantial cohort of young adults in early spring, evidenced by new small-sized components in October for males (mean = 23.83 mm) and November for females (mean = 23.51 mm) (Figure 3). Individuals of this cohort became predominant by January, creating a significant population turnover with smaller size classes comprising 72% of males (mean = 36.86 mm) and 93% of females (mean = 33.56 mm).

An additional incorporation of young adults was detected in February for both sexes, with new small-sized components (20.87 ± 3.00 mm in males, 20.28 ± 1.64 mm in females) that contributed substantially to the population in subsequent months (Figure 3). Two winter modal components of young adults were observed with minimal impact on the overall population structure: a male component in August (CW = 15.87 ± 2.49 mm) and a female component in June (CW = 20.46 ± 2.53 mm). The temporal pattern of juvenile abundance showed an increase from November (1.1 individuals) to a peak in March (17.4 individuals), almost mirroring seasonal temperature oscillations (Figure 4).

Accelerated growth was evident during spring and summer months (October–March), as indicated by the increasing mean sizes of modal components showing rapid size increases during this period (Figure 3).

### 3.3. SM_50_ and Ovigerous Female Seasonality

The weighted logistic regression model revealed a significant relationship between the proportion of ovigerous female crabs and CW (GLM, z = 31.44, *p* < 0.001). The carapace width at which 50% of female crabs were ovigerous (SM_50_) was estimated to be 61.48 mm (Figure 5).

Ovigerous females were present in the intertidal zone throughout nearly the entire year, underscoring the species’ prolonged reproductive activity in this region. However, the proportion of egg-bearing females exhibited a pronounced seasonal pattern, with a coefficient of variation of 1.27 (127%) for the weighted annual proportions. The primary reproductive peak occurred during May–July, accounting for 62% of the annual reproductive effort, with July alone representing 38.9%. A secondary reproductive period was observed from September to December (spring to early summer), which collectively accounted for 30.6% of the annual reproductive effort. Late summer to early autumn months (February–April) showed substantially reduced or absent reproductive activity, with March having no ovigerous females (Figure 6).

## 4. Discussion

This study provides baseline demographic and life-history data for the newly established population of the invasive green crab (*C. maenas*) in the San Matías Gulf, at the northernmost edge of its known range in the southwestern Atlantic. The overall crab density of 0.42 ind m^−2^ showed significant temporal and spatial variations, with lower densities in winter and higher densities during warmer months. Our findings revealed a pronounced male-biased sex ratio, with males exhibiting 79% higher density than females and attaining significantly larger sizes. Green crabs showed a significant population turnover, with a substantial cohort of young adults emerging in early spring that became predominant by January, followed by an additional recruitment in February. Modal progression analyses revealed accelerated growth during spring and summer months (October–March). Moreover, we present the first estimate of the size at which 50% of females are ovigerous for the southwestern Atlantic coast. Ovigerous females were observed in the intertidal zone throughout most of the year, though with significant temporal variability and a distinct bimodal pattern, demonstrating the species’ extended yet seasonally structured reproductive period and its remarkable adaptability to local environmental conditions.

### 4.1. Density and Sex Ratio

The overall density of *C. maenas* in the San Matías Gulf (0.42 ind m^−2^) falls within the lower range reported for different populations of the green crab globally (Table 2), providing valuable baseline data for this recently detected population, first recorded in 2022 [18]. Seasonal fluctuations were evident, with the lowest densities recorded during winter and a steady increase observed during the warmer months. These trends align with seasonal reproductive and molting cycles documented in other *C. maenas* populations, supporting previous research that water temperature influences crab distribution patterns, with offshore migration during colder months and inshore movement during warmer periods [35].

By comparison, established populations of *C. maenas* report a mean density of 21.8 ± 34.5 crabs·m^−2^ across 16 studies (Table 2). However, the methodological diversity across studies complicates direct density comparisons. Our hand-collection approach aligns well with other hand-collection studies (1.56 ± 0.7 ind m^−2^, n = 8 studies, Appendix A), suggesting that our findings are consistent with similar methodologies, despite the lower values. Even more, between these studies, intertidal estimates fall between 0.24 and 1.29 ind m^−2^ [36]. In contrast, studies using other sampling methods are scarce and show considerable variability in their estimates. Techniques such as substrate sieving (45.56 ± 20.95 ind m^−2^, n = 5; [37,38]) and trawl-net sampling (52.75 ind m^−2^, n = 2; [39]) yield substantially higher densities, as they often include juveniles and larvae. Similarly, trapping methods (3.10 ind m^−2^, n = 1; [40]) selectively capture actively foraging individuals, while mark–recapture methods can be compromised by tag loss during molting [5]. These inherent limitations emphasize the need for standardized methodologies to facilitate more meaningful comparisons across invaded regions.

Our results revealed a strongly male-biased sex ratio (1.50 males per female), a pattern commonly observed in *C. maenas* populations worldwide (e.g., [41,42,43,44]; Table 2 and Appendix A), as well as in populations of native crab species (e.g., [45,46,47]). This bias is often attributed to differences in habitat use, behavior, and physiological requirements between sexes. Females, for instance, migrate to subtidal zones during the breeding season to release larvae in more stable conditions, resulting in reduced numbers in intertidal areas during low tide sampling periods [48]. Alternatively, the male bias could reflect sampling bias or behavioral differences, as males are generally more active and exploratory, increasing their likelihood of being captured [49].

### 4.2. Size

In line with global patterns for *C. maenas*, males in the San Matías Gulf attained significantly larger carapace widths than females (mean: 40.11 mm vs. 36.79 mm; Table 2, Figure 7). Interestingly, the maximum sizes recorded in our study (72.1 mm for males and 64 mm for females) fall within the lower range reported for other populations documented globally (Table 2). Several non-exclusive hypotheses could explain this pattern.

First, our sampling methodology may have contributed to the observed size distribution. The hand-collection approach in intertidal zones typically yields smaller individuals than trapping methods, which target larger crabs in subtidal areas (77.21 ± 4.1 mm vs. 88.94 ± 10.3 mm mean carapace width for males across studies; Appendix A). This pattern aligns with documented ontogenetic habitat partitioning in *C. maenas*, where smaller individuals dominate intertidal zones, while larger crabs migrate to deeper subtidal areas [50]. Our study was conducted exclusively in the intertidal zone, which likely limited our ability to sample the largest individuals that might inhabit deeper waters. However, spatial segregation alone cannot fully explain our findings, as our maximum recorded sizes remain below the averages reported even among other hand-collection studies, suggesting additional factors are influencing the size distribution in this population.

The relatively smaller sizes we observed likely reflect traits characteristic of newly established populations experiencing strong *r*-selection pressures. At the leading edge of a species’ range or during early invasion stages, selective forces typically favor traits that enhance rapid population growth and survival, including reduced age at maturity, accelerated growth rates, and increased energy allocation to reproduction rather than somatic growth [2,51,52,53,54]. In *C. maenas*, size is an important determinant in competitive interactions between males [55]. Low population density may reduce intraspecific competition, potentially allowing for successful reproduction at smaller sizes compared to dense populations where larger individuals have competitive advantages. These adaptations enable individuals to reach reproductive size sooner, thereby shortening generation times and accelerating population expansion [56,57]. Such life-history strategies are particularly advantageous for invasive species, facilitating both primary invasion and secondary spread [58,59].

Lastly, the presence of hybrids between *C. maenas* and *C. aestuarii*, coupled with the general lower sizes reported for this species [60], could also explain the intermediate size ranges observed in this study. Hybridization has been documented in regions such as Japan, where both species co-occur [61], and could produce phenotypes with traits from both parent species [3,62,63], potentially influencing growth patterns and size distributions. Recent molecular evidence suggests *C. aestuarii*’s presence in the nearby Nuevo Gulf [64]; therefore, the possibility of hybrids in the San Matías Gulf cannot be ruled out and warrants further investigation. However, we consider this explanation less likely for several reasons. Our morphological assessment using established criteria [65] indicated the unique presence of *C. maenas*. Furthermore, genetic analyses by Darling et al. [66] determined that *Carcinus* populations in Argentina likely derived from *C. maenas* populations introduced from Australia. Although Cordone et al. [64] detected sequences from both species in metabarcoding analyses of gut contents, the authors acknowledged the limitations of metabarcoding in distinguishing actual prey items, and neither *C. maenas* nor *C. aestuarii* were visually identified in their study. Based on this evidence, we conclude that the San Matías Gulf population most likely represents *C. maenas*, with the observed size patterns primarily reflecting invasion dynamics and the sampling methodology rather than hybridization effects.

### 4.3. Population Size Structure

The size-structure analysis of the green crab population revealed distinct bimodal size distributions in both sexes, reflecting a structured population with clear cohort separation. Unlike males, which consistently displayed two size classes throughout most of the year (with the exception of August), females exhibited greater variability in modal structure, ranging from one to three size classes depending on the month. This sex-based difference in population structure dynamics aligns with findings from other regions where males and females exhibit different growth and molting patterns [67].

Our temporal analysis identified two significant recruitment of young adult events. The first occurred in early spring (October–November), when substantial cohorts of young adults emerged for both sexes. By January, these cohorts became predominant, creating a significant population turnover with smaller size classes comprising the majority of the population. A second recruitment event was detected in February, with new small-sized components contributing substantially to the population in subsequent months. The temporal pattern of juvenile abundance, which increased from November to a peak in March, likely captures this second recruitment pulse. Two additional winter modal components were observed with minimal impact on the overall population structure, suggesting limited winter recruitment. This bimodal pattern of recruitment aligns with observations from other populations of *C. maenas* across its native and invasive ranges. In European populations, refs. [67,68] documented one or two main recruitment periods annually. Similarly, ref. [69] found that in non-native Atlantic coast populations of North America, recruitment typically occurs in multiple pulses, with two principal events coinciding with water temperature cycles. In Argentina, Vinuesa [21] also reported comparable recruitment patterns, with a primary event in spring and a secondary one in late summer. This pattern appears to be relatively consistent across the species’ global distribution [5,61].

Notorious growth was evident during spring and summer months (October–March), as indicated by rapidly increasing modal sizes during this period. This accelerated growth coincides with warmer water temperatures, conditions that, coupled with an increased food availability, favor ecdysis in decapod crustaceans [70]. The seasonal growth pattern observed in our population is consistent with other studies of *C. maenas* that document concentrated molting periods during warmer months [61,71,72]. This growth pattern indicates a concentrated molting period, which has significant implications for reproduction, since female green crabs can only be fertilized during the soft-carapace period immediately following ecdysis [5,21].

### 4.4. SM_50_ and Ovigerous Females

The size at which 50% of females were ovigerous (SM_50_) in this study was 61.48 mm. This value is larger compared to sizes at maturity estimates from other *C. maenas* populations (e.g., [13,33,48]; see Table 2 and Appendix A), where values typically range from 28.9 mm to 49.96 mm. While previous studies primarily relied on macroscopical and histological techniques to determine gonadal and physiological maturity, our SM_50_ estimate was based on the proportion of females observed carrying eggs [33,47]. This method inherently reflects the end of the reproductive process—when fertilized females are bearing eggs—rather than the onset of physiological maturity. As a result, the actual size at physiological maturity is likely smaller than our SM_50_ estimate. Unfortunately, no studies exist on SM50 based on beared females for other *C. maneas* populations.

The duration of the reproductive period in green crabs appears to be influenced by temperature, with ovigerous females reported for three months in colder regions and up to eight months in warmer locations [13]. This pattern may also apply to the southwestern Atlantic coast, where in colder areas, ovigerous females are typically observed between May and September [21]. In contrast, in the San Matias Gulf, we detected ovigerous females in the intertidal zone nearly year-round. Similar extended reproductive periods have been documented in the species’ native range [67,72], suggesting that favorable environmental conditions in this region could support prolonged reproductive activity. While some studies have proposed that prolonged brooding might reflect lower reproductive success due to environmental stress, as reduced salinity, lower temperatures, or mild disturbances can delay larval release in *C. maenas* [5,73], this explanation seems unlikely for our study population. In the San Matias Gulf, fluctuations in salinity and temperature remain well within the species’ tolerance range, making it improbable that abiotic factors limit population establishment or reproductive success at this stage.

Although ovigerous females were present throughout most of the year, our results revealed a pronounced seasonal pattern, with the highest proportions observed between May and July. The observed low proportion of ovigerous females in August, coupled with the subsequent emergence of a substantial young adult cohort in October, strongly suggest that larval release predominantly occurs during August. This timing is consistent with the bimodal size distributions in both sexes and discrete recruitment peaks. Temporal growth patterns further support this seasonal reproductive pattern. However, the extended period during which ovigerous females can be found suggests considerable reproductive flexibility, which may enhance recruitment success by increasing the probability of successful larval release under optimal environmental conditions. This adaptability could be particularly advantageous for newly established invasive populations, allowing them to capitalize on favorable conditions rather than being constrained to a narrow breeding window, ultimately facilitating population growth and range expansion. Year-round systematic larval sampling is required to definitively confirm this reproductive pattern.

## 5. Conclusions

This study provides baseline demographic and life-history data for the newly established green crab population at the northernmost edge of its known range in the southwestern Atlantic. By employing a direct counting approach in the intertidal zone, we captured an initial snapshot of key population traits. The population exhibited a male-biased sex ratio, relatively low density, and relatively smaller body sizes, suggesting it is in an early stage of establishment. Ovigerous females showed a clear seasonal pattern, but their year-round presence indicates a potential extended reproductive period, which may enhance recruitment success and promote population expansion. The long-term monitoring of population density, size structure, and reproductive patterns, along with genetic analyses to confirm species identity among all the Patagonian green crab populations, will be crucial for understanding the dynamics of this invasion. As post-establishment management actions are often ineffective [74], preventing additional introductions emerges as the most resource-efficient method to limit the green crab’s ongoing expansion [75]. This precautionary approach is especially critical in marine ecosystems, where extensive connectivity across broad spatial scales renders the eradication of established non-native species.

## Figures and Tables

**Figure 2 biology-14-00480-f002:**
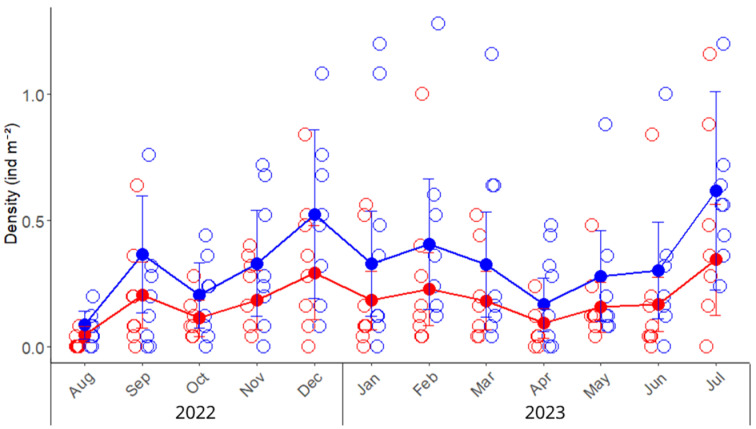
Monthly density (ind m^−2^) variation of the invasive green crab *Carcinus maenas* in the intertidal zone of the San Matías Gulf (Las Grutas, Río Negro, Argentina), from August 2022 to July 2023. Empty circles represent observed densities from sampling quadrats (25 m^2^ each) placed in the upper and lower intertidal zones, while solid points, lines, and error bars indicate predicted means and standard errors derived from the mixed model. Red and blue colors indicate the densities of females and males, respectively.

**Figure 3 biology-14-00480-f003:**
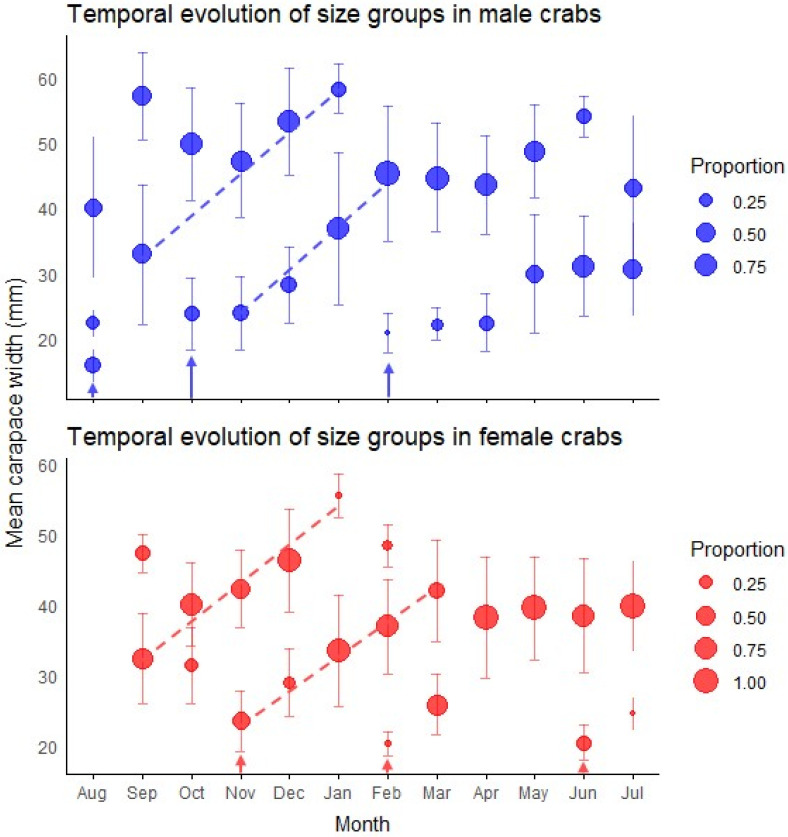
Temporal patterns of size structure in green crab populations. The plots display the monthly progression of size groups identified by mixture model analysis for both male (top panel) and female (bottom panel) crabs throughout the year. Circle size represents the proportion of individuals within each size group, while vertical bars indicate standard deviations. Dashed lines track the most conspicuous growth progression of modal size classes over time. Arrows indicate recruitment events where new cohorts appear in the population. Monthly size-class frequency distributions are presented in Figure A4 and Figure A5 and their associated statistics are presented in Table A3 and Table A4.

**Figure 4 biology-14-00480-f004:**
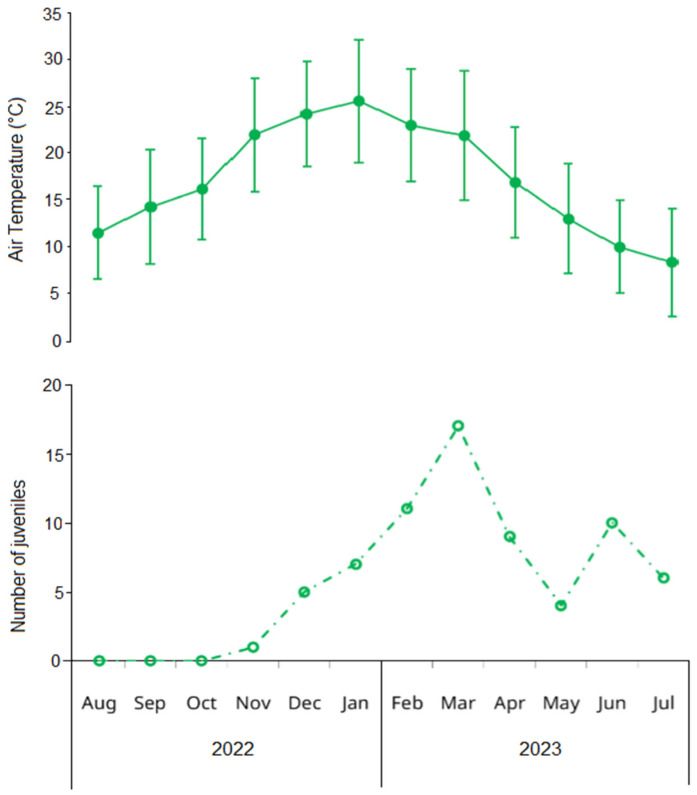
Upper panel: mean (solid points) and standard deviation (error bars) of monthly air temperature in Las Grutas (Río Negro, Argentina) obtained from the Saint Exupéry airport climatologic station. Lower panel: monthly variation in the number of juveniles of the invasive green crab *Carcinus maenas* in the intertidal zone of San Matías Gulf (Las Grutas, Río Negro, Argentina), from August 2022 to July 2023. Juveniles are defined as small individuals (<15 mm carapace width) whose sex could not be determined through visual examination of the abdomen.

**Figure 5 biology-14-00480-f005:**
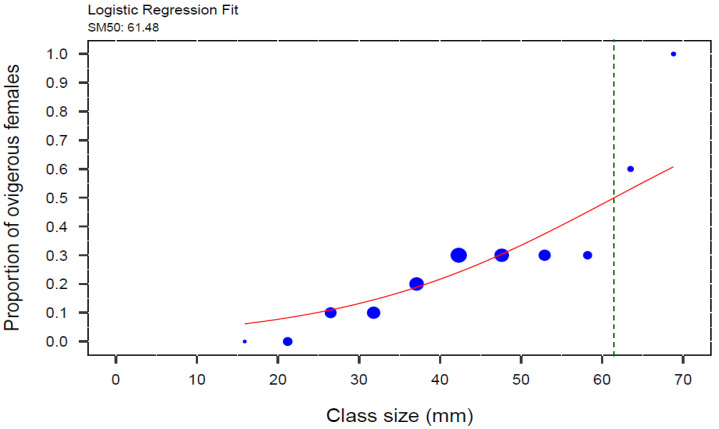
Weighted logistic regression of proportion of ovigerous females to carapace width (CW) of the invasive green crab (*Carcinus maenas*) in the intertidal zone of the San Matías Gulf, Argentina, from August 2022 to July 2023. SM_50_ for female crabs was estimated at the inflection point of the curve, which corresponds to 61.48 mm CW (N = 472), indicated by the green dotted line. The blue circles represent the proportional size of the points, scaled by the total number of individuals in each size class, grouped in 5.3 mm size-class intervals.

**Figure 6 biology-14-00480-f006:**
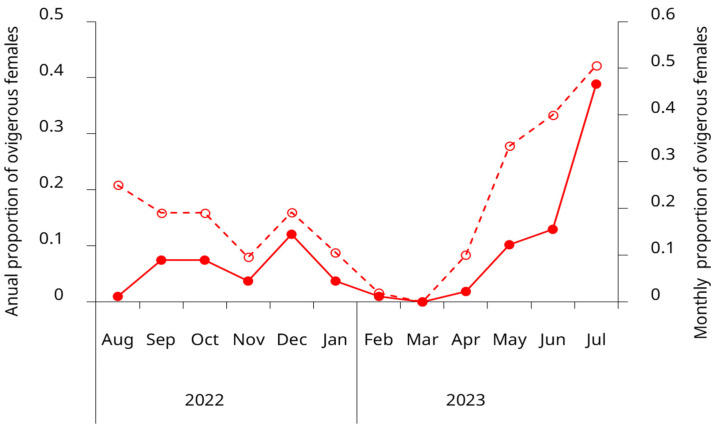
Seasonal variation in the proportion of ovigerous females of the invasive green crab (*Carcinus maenas*) in the intertidal zone of the San Matías Gulf, Argentina, from August 2022 to July 2023. Open circles and the dashed line show the monthly proportion of ovigerous females (left *y*-axis). Solid circles and the solid line represent the annual distribution of ovigerous females by month, showing the proportion of the year’s total ovigerous females observed each month (right *y*-axis). Data represent pooled observations from both upper and lower intertidal zones.

**Figure 7 biology-14-00480-f007:**
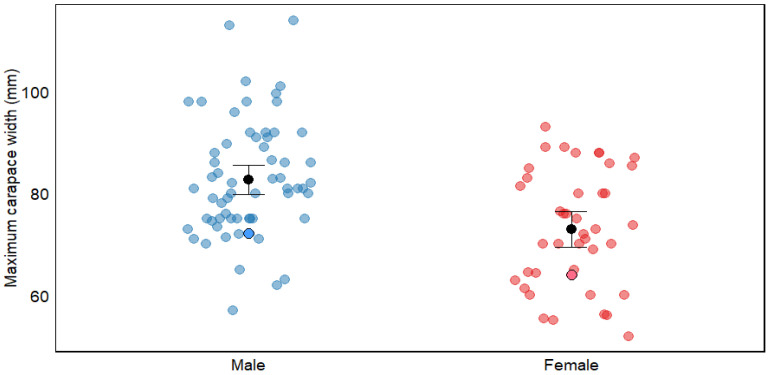
Maximum carapace width (mm) of *Carcinus maenas* by sex, showing comparative data between males and females. Blue dots represent males and red dots represent females, with data from both bibliographic sources (pale dots) and this study (dots with border lines). Black dots with error bars represent means ± 95% confidence interval. The figure illustrates sexual dimorphism in carapace width within this species. For detailed information on size and references, see Appendix A.

**Table 1 biology-14-00480-t001:** Summary of means (±SE) and statistical significance of sex effect on density (ind m^−2^) and size (carapace width, mm) in response to sex difference of the invasive green crab *Carcinus maenas*, from the intertidal area of the San Matías Gulf, Argentina. Number of observations are indicated between parentheses. Significant values of each effect are shown in bold. Complete mixed-model results including random effects can be found in Appendix B (Table A1).

Variables	Sex		Sex Effect	Sex Comparison	Effect Size
	Females (F)	Males (M)			
Density	0.15 ± 0.06 (96)	0.26 ± 0.11 (96)	*X*^2^_1_ = 21.05;	F vs. M	−4.59
			*p* **< 0.001**		
Size	36.79 ± 1.02 (472)	40.11 ± 1.24 (709)	*X*^2^_1_ = 22.25;	F vs. M	−4.72
			*p* **< 0.001**		

**Table 2 biology-14-00480-t002:** Morphometric and population parameters of *Carcinus maenas* obtained through bibliographic review. Data are presented as mean ± SD (n) and range (min–max) in parentheses. The complete dataset is available in Appendix A.

Sex	Maximum Carapace Width (mm)	Sex Ratio (M–F)	Density (Crabs m^−2^)	SM_50_
Male	82.7 ± 11.5 (62) (57.0–114.0)	1.8 ± 1.6 (36) (0.4–8.2)	21.8 ± 34.5 (16) (0.1–101.2)	38.3 ± 14.0 (6) (28.9–49.96)
Female	72.9 ± 11.4 (42) (52.0–93.0)			

## Data Availability

The data supporting the findings of this study are available in the Zenodo digital repository at https://doi.org/10.5281/zenodo.15278353.

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
