# Peer review of "Moving Northwards: Life-History Traits of the Invasive Green Crab (Carcinus maenas) Expanding into the Southwestern Atlantic"

_biology, 2025, doi:10.3390/biology14050480_

Round 1

Reviewer 1 Report

Comments and Suggestions for Authors

I think that the manuscript titled Moving northwards: Invasive Green Crab (Carcinus spp.) Expands into the Southwestern Atlantic, is of great interest since it provides data on an invasive species in the study area. The article is well presented and I think it can be published. However, I have some specific comments:

-In the discussion section the authors state that: “Ovigerous females were observed in the intertidal zone throughout most of the year, despite a distinct seasonal reproductive pattern, indicating the species’ extended reproductive activity and its remarkable adaptability to local environmental conditions.”

I think that they could compare these results with those obtained in other studies and expand the discussion. Is the length of the reproductive period correlated with latitude (temperature) as in many other species? Do ovigerous females also appear throughout the year in other geographic locations?

Minor comments:

-Figure 1: I think the authors should increase the font size that appears on the map (references and coordinates).

-Materials and Methods: It is better to add the definition of carapace width (CW); for example: distance between the tips of the fifth teeth on the anterolateral margin of the carapace.

In the future, I encourage the authors to carry out a genetic study to clarify the origin of the population of this locality (C. maenas, C. aestuari, C. maenas X C. aestuari, native origin...).

Reviewer 2 Report

Comments and Suggestions for Authors

The manuscript presents evidence of the strategies used by an invasive species, documenting its expansion in the Southwest Atlantic, based on population structure. The document is clear, precise, and organized.
I suggest that in the small box of Fig. 1, you add the name of the Atlantic Ocean. In the large box, I suggest that you add the names of the Bays and Gulfs. I also suggest that you add a figure of the orientation towards the North.

Reviewer 3 Report

Comments and Suggestions for Authors

The work “Moving northwards: Invasive Green Crab (Carcinus spp.) Expands into the Southwestern Atlantic” by M. M. Baigorria et al. focuses on a green crab, that is a well studied invasive species and is considered as one of the world’s 100 worst invasive species. Understanding its adaptation strategies, population structure and spreading is key to predict its further distribution, possible impact levels and select strategies for management. Although these issues are highlighted in the introduction of the paper, the information given in this work does not substantially contribute to their understanding.

In particular:

Introduction:

A large proportion of the introduction discusses the different factors contributing to the invasiveness of a species, susceptibility of the invaded environment and the impact of invasive species and crabs in particular (first 3 paragraphs), while no data on these issues is given in the results or discussion section of the manuscript. No data on the niche availability (disturbance levels) or impact of these crabs in the studied area is given. Probably it would be better to give less focus to these issues in the introduction. One or two phrases would be sufficient.

A second issue largely discussed in the introduction in the need of genetic testing to differentiate between two closely related green crab species. This work gives no data on the genetic makeup of the analyzed specimens. Although, the two species can have different sizes, they are still analyzed as a bulk in this work and inconsistencies in average maximum sizes are attributed to the possibility of their coexistence and hybridization. Throughout the text the specimens collected and analyzed are often referred to as “this species” – as in assuming it is one species, while in other instances in plural. Once again, if the data provided in the manuscript does not provide insight into the species composition, it would probably be sufficient to mention this issue only in couple of phrases in the introduction.

The data provided in this work is size composition and presence of ovigerous females. However, there is no information about the general biology of these crabs in the introduction. What are their growth patterns, how does that reflect their stage of introduction in a new territory.

Furthermore, history of their findings is given along the Atlantic shore of South America and their northwards expansion. No data is given on the temperature, salinity tolerance of the species and how these and other abiotic factors can limit their expansion. Are the new findings within their tolerance levels or marginal? Such information is directly related to the size composition and the expansion of an invader, which seems to be the main focus of the manuscript. Although no clear hypothesis is given in the introduction. The authors claim to “By comparing these traits to those reported for other Carcinus spp. populations at varying stages of establishment, this study aims to provide valuable insights into how invasive species establish and modify their life history traits across invaded environments.” – although no varying stages of establishment are specified in either results or discussion sections, it is unclear what life history traits are modified.

In the Methods.

Very little information is given on the actual method of collection which makes it impossible to compare to other studies. How were the crabs collected? By hand? Running after them? Do they borrow or hide in crevices? Especially molted females and juveniles that hide. If the crabs were collected during low tide, could have they migrated to subliteral zone? And in different proportions between sexes? Strong men remaining in the open. How about soft crabs? Especially females that have molted? All of this issues are mentioned in the discussion but no information of how they were compensated for is given in the methods.

It is hard to judge how accurate is the calculated density of crabs when the reader does not know if all crabs were collected in the studied area. Throughout the manuscript the authors mention “population density” and “population size structure” however, how representative are sampled crabs of the actual population is unclear. Further in the discussion it is mentioned that other worked use different methods of collection and have different results, but no description or comparison of these methods is given, therefore it is hard to judge whether the results are accurate or even comparable.

Also:

Do these crabs have difference in life histories based on sex and size an how did the method of collection account for that?

Did you find any females post larval release? Were there months with higher proportion of such females?

Why did you release them if they are such damaging invasive species and it would have been interesting to look at their reproductive characteristics such as embryo and gonads development stages. These characteristics would give a better understanding of the reproductive potential of crabs and their life history traits.

Line 133-134: What are the two distinct adjacent zones? What is their distinction if the slope is gentle? How were they differentiated?

Line 145: Why were crabs released into the wild if they are in the top 100 worst invasive species that have such an adverse effect on the environment? This is not a criticism as such, as a mere curiosity. If they are invasive species, could their invaded habitat be described as “natural habitat”?

Line 148-149. Using Sturges' formula for class intervals may be handy in many circumstances to assign appropriate bin sizes, however it is a bit problematic to then further refer to these bins as size classes. Crabs are molting organisms and grow in impulses. It is probably more appropriate to use “size classes” for the modes/means and SD that result from the growth impulses rather than to apply a uniform 5.3 class intervals. For example one can used mixture model analysis to identify individual size groups in a population. Such analysis would give a more realistic size groups of a molting species population size structure. In addition such size groups can give a better understanding of the most dominant size groups, ei, show whether the population is young or old or all size and ages are present, which could shed light on the progress of the invasion. This is much more visually apparent in a histogram. Therefore, I believe that the figures could be clearer in a histogram form.

Results

Better start with how many crabs were analyzed and then talk about the size and density differences

What is the density of juveniles? Only females and males’ densities are given.

 The whole section of size structure is unclear. See comments in the methods section. I would highly recommend to look at the size structure through histograms, look and compare different size instars and compare them to those given in other areas both natural and invaded. Mixture model analysis would give different size groups. These may shed light on the possible species composition if there are known size differences. The number and sizes of juveniles are very important in the newly invaded populations as they show successful recruitment. I am sure the authors have good data to show that.

The data was collected monthly. However this is not reflected in the results. Graphs of densities sex composition, proportion of ovigerous females by month would be highly informative to discuss their actual like strategies in the tidal zone (probably mostly reflecting their migration.

And most importantly why was SM50 parameter chosen, when further in the discussion section authors admit that “While previous studies primarily relied on macroscopical and histological techniques to determine gonadal and physiological maturity (Table 2), our study estimated SM50 based on the proportion of females observed in the ovigerous condition. As a result, it is likely that the size at which 50% of females achieve physiological maturity is smaller than the SM50 estimate reported  here.” Just because females carry eggs in different months does not necessarily reflect highr reproductive potential. It does not directly mean that they release more than one batch per year, may be they just carry them longer. And also no information I given on the development stages of the embryos or their quality. It is possible that eggs can be mostly damaged and therefore not released for a long time. Therefore, longer period of ovigerous females can actually be due to lower reproductive capability in the new area.

Discussion section is mostly focused on the issues with the method of collection and analysis that I also raise above, however give no clear answer to the comparative power and reliability of the given results. After reading the discussion section, I was left with an understanding that neither the methods nor the results are comparative to other works. And very little literature is cited.

Given all of that, I still believe that the data collected by the authors can be highly informative and can contribute to better understand the expansion of green crabs in the southwestern Atlantic. I highly recommend to reconsider the data analysis and representation. It seems that so much more can be gained from monthly collection of samples, than a bulk size cloud of males and females. I therefore recommend to rethink and rewrite this work completely.

Reviewer 4 Report

Comments and Suggestions for Authors

The manuscript investigated the sex ratio, population density, size distribution and the proportion of ovigerous females in Carcinus spp. population in the San Matías Gulf. The MS reported a newly identified invaded site for Carcinus spp., and contributed to understanding the population's life history traits. However, molecular analyses are absent in the study, leading to unreliable conclusions. I strongly suggest the authors run a follow-up experiment on genetic makeup. Besides, there are some issues that need to be revised.

  1. Line 18, I don't think the results are enough to understand how the population might grow and spread in the future.
  2. The Carcinus spp. population in this study may include Carcinus aestuarii, C. maenas and hybrids between the two. The article repeatedly mentions the morphological similarity between C. aestuarii and C. maenas, emphasizing the importance of molecular identification (line 84-88, line 97-98, line 267, line 319-321). However, the study did not differentiate these two species in its experimental design. If species identification had been conducted first, followed by separate statistical analyses of their characteristics, would the results have been different?
  3. Line 42, the assertion regarding management applications represents a common limitation in biological invasion research, where in practical management recommendations are not derived from the collected data. To strengthen this section, either the statement should be eliminated or specific management strategies should be proposed in the discussion, grounded in the study's findings.
  4. I think the content of section 2.3 should be merged into section 2.4.
  5. Subheadings should be added in the results section.
  6. Does "the early stage of population establishment" imply that the population is experiencing a bottleneck effect? I believe that assessing the status of an invasive population solely based on population density is not reasonable. Molecular data, particularly the genetic diversity of the population, should also be considered.
  7. The study site represents the northernmost recorded range of this species. Does the relatively low population density observed in this study relate to temperature? Has the temperature range for survival of this species been studied?
  8. Line 219, vague phrase
  9. Do these traits—male-biased sex ratio, larger male individuals, and year-round reproduction—share similarities with other non-invasive crab species?

Round 2

Reviewer 3 Report

Comments and Suggestions for Authors

The authors have thoroughly revised their manuscript, adding necessary information in the introduction, to set the scene, detailed and clear methods and extensive results. At this point I do not have any major comments on the quality of the research and its report. There is only one small suggestion:

Fig. 2. Very hard to see the difference between light and solid points. May be better make one empty and one solid for better contrast?

However, I am still unsure that this work is fundamental and interesting enough for a first quartile journal such as Biology. It is definitely useful and important information that should be published. However in my opinion it is of local importance. Even though the authors report that this is the most northern population of green crabs, it is still within suitable abiotic conditions, and seems to be a natural expansion of the invasive population. However, I am not qualified to judge what is sufficient scientific value for the journal and I am sure the scientific editor will make a better decision.

Otherwise, in this format the manuscript is of sound scientific value and quality and can be published.

Author Response

Comment 1: Fig. 2. Very hard to see the difference between light and solid points. May be better make one empty and one solid for better contrast?

Response 1: Done

Reviewer 4 Report

Comments and Suggestions for Authors

I thank the authors for their attentive attention to my comments. The presented version of the manuscript is more enriched now and can be published after a minor revision.

  1. It would be better to indicate the location noun “intertidal zone” in both Simple Summary and Abstract.
  2. In line 32, the “size” should be indicated clearly as carapace width.
  3. In consideration of the main work, I think it would be better to modify the title using demographic characteristics, population, etc.
  4. Figure 7 is missing.
  5. Other errors, for example, to italicize “Study” in line 99 and the subtitle in line 206, to normalize the top row of Table 1, to replace “ind m2” by “ind m-2” in Figure2, to capitalize the first letter of “individual size” in line 174.

Author Response

Comment 1: It would be better to indicate the location noun “intertidal zone” in both Simple Summary and Abstract.

Response 1: Done.

Comment 2: In line 32, the “size” should be indicated clearly as carapace width.

Response 2: Done.

Comment 3: In consideration of the main work, I think it would be better to modify the title using demographic characteristics, population, etc.

Response 3: Done. We've modified the title. Now it is: "Moving northwards: Life-History traits of the Invasive Green Crab (Carcinus maenas) expanding into the Southwestern Atlantic"

Comment 4: Figure 7 is missing.

Response 4: Rigth, we´ve corrected that.

Comment 5: Other errors, for example, to italicize “Study” in line 99 and the subtitle in line 206, to normalize the top row of Table 1, to replace “ind m2” by “ind m-2” in Figure2, to capitalize the first letter of “individual size” in line 174.

Response 5:: Done